# Restorative Practice and Therapeutic Jurisprudence in Court: A Case Study of Teesside Community Court

**Susie Atherton**

School of Natural and Social Sciences, University of Gloucestershire, Cheltenham GL50 2RH, UK; satherton3@glos.ac.uk

**Abstract:** This article examines the contribution of restorative practice and therapeutic jurisprudence in community courts, which have adopted a problem-solving approach. Through interviews with stakeholders, it explores the implementation of the community court model in Teesside. This work draws from a broader study in Middlesbrough, which adopted a case study design, to profile the local community and to present experiences of community justice, including the community court. For this article, there is a specific focus on the data collected from those working in the community court and in partnership with it. The findings demonstrate both the benefits and challenges of courts adopting problem-solving approaches. There was clear support among magistrates who recognised the value of doing justice differently, to more effectively dealing with re-offending. Among all participants, positive experiences and outcomes were reported, alongside acknowledgement of the logistical and political challenges associated with implementing innovations in criminal justice. This included negative media representations and a lack of investment to sustain the change in practice. Participants across the sample emphasised the importance of adopting a different ethos, aligning with restorative practice and therapeutic jurisprudence and shifting away from adversarial approaches to present a more effective response to the problem of crime.

**Keywords:** justice; community court; problem-solving; restorative practice; therapeutic jurisprudence

## 1. Introduction

This article examines the implementation of the community court introduced in Teesside in 2007, focusing on the alignment between problem solving approaches, restorative practice and therapeutic jurisprudence. It explores the rationale for adopting different practices within court settings, examining the benefits and challenges of this, in a community which has experienced high levels of crime and anti-social behaviour, alongside significant economic decline. Given that the model adopted by the Teesside Community Court is no longer represented as active in the courts database, this is a reflective account of changes in practice adopted by the court staff in 2007. In England and Wales, community courts were piloted in 10 areas, where a 'problem-solving approach' was adopted, to replicate the model still in use in the USA, for example the Red Hook Community Justice Centre in Brooklyn, New York. Prior to this a community justice centre (CJC) was built in north Liverpool, to replicate the Red Hook CJC. The community court model was not rolled out further across England and Wales after this pilot and now remains as provision in some court areas, such as Plymouth Community Court working with the Community Advice and Support Services (CASS) (Annison et al. 2013). However, it is useful to reflect on the implementation of the community court model and to examine the role of restorative practice and therapeutic jurisprudence (Ward 2014, p. 2) in offering more 'socially meaningful' justice (Donoghue 2014, p. 141). In addition, a rationale for change to the court system was clear from Bowen and Whitehead's (2013) report highlighting issues with delays, discontinued cases and the revolving door of justice, which are problems that remain today.

The history of community courts and community justice centres in the UK is one of shifting priorities in government and also reflects resistance to changing entrenched notions of 'justice'. Therefore, this approach remains peripheral to mainstream criminal justice practice, and consequently has suffered from a lack of investment and support from central government. This has occurred despite evidence of the value of community courts, specifically with the use of sentencing and the formality of the court setting to effectively address the causes of crime and monitor compliance with sentences (Karp and Clear 2000; Mair and Millings 2011). This model views the courtroom as a place in which dialogue between the judge, defendant and legal representatives occurs to establish what led to offending behaviour and, therefore, what could be put in place to prevent re-offending (Wolf 2007; Llewellyn-Thomas and Prior 2007). In conjunction with their role as passing the sentence, magistrates work with other court staff to signpost defendants to services to deal with a range of problems such as debt, addiction, housing issues and help with employment (Llewellyn-Thomas and Prior 2007). Donoghue (2014) describes this as *individualized* sentencing, informed by the circumstances of the defendant in order to have a more effective outcome in preventing re-offending.

A report by the House of Commons Justice Committee (HoCJC 2010) examined broader issues relating to the rationale for reform in the criminal justice system (CJS) and challenges to this. The report suggested 'prehabilitation' as a more effective and humane response, to tackle broader issues contributing to offending and to disregard criticism represented by media reporting of new approaches to justice. Specifically, this change aimed to focus on a 'locally responsive system of community sentences' (HoCJC 2010, p. 8) which would also require partnership working beyond the CJS to enable effective problem solving. This emphasis on local response aligns with Rawls' (1971) assertion that justice should be a 'stabilising force' to reinforce the 'bonds of civic friendship' (p. 5) in communities, given the disruptive impact of crime on social cohesion and trust. The HoCJC (2010) report also advocates for problem solving approaches, which are the key feature of community courts, and align with restorative practice. This practice, described by Quinney (2000) as 'peace-making criminology' focuses on repairing harm and resolving conflict, presenting a means by which to achieve Rawls' (1971) view that justice can stabilise communities.

However, while problem solving courts represented a genuine attempt to move away from justice as an adversarial contest (Karp and Clear 2000), it is clear that doing justice differently faces a number of challenges. For example, Giddens (1991) argued that community justice has too often been perceived as 'utopian realism'. It is also suggested that while community justice offers viable alternatives to punitive and repressive sanctions, this has occurred in a climate of 'denial' about the broader causes and impact of crime (Garland 2001). It can be argued that innovations in justice have attempted to address these concerns, specifically to better acknowledge the broader causes of crime, as found with problem solving and specialist courts (Donoghue 2014). In addition, in England and Wales there is evidence of the effectiveness of problem-solving approaches in courts (Mair and Millings 2011; Annison et al. 2013), restorative practice to resolve civil and criminal cases (Sullivan and Tift 2001; Johnstone 2013) and adopting a therapeutic ethos within legal processes (Ward 2014). While it is understandable that the notion for such practice to form the key features and functions of the CJS is perceived as 'utopian' (Giddens 1991), this should not deter reflection on and examination of alternatives to adversarial processes.

This article focuses on the contribution of restorative practice and therapeutic jurisprudence, as used in problem solving approaches adopted by the community court model. The data used in this article is from semi-structured individual interviews with court staff and those working in partnership with the community court. The data is analysed in the context of identifying where both restorative practice and therapeutic jurisprudence aligns with problem solving approaches, along with reflection on the challenges for implementing alternative forms of justice.

## 2. The Problem-Solving Court Model and Restorative Practice

The Red Hook Community Justice Centre in Brooklyn, New York, delivered justice and provided a range of support services for local residents, such as peer learning, social activities for young people, drug and alcohol treatment and access to education and training for local residents (Llewellyn-Thomas and Prior 2007). An evaluation of this court demonstrated it had met its aims of transforming the local community, from a district described as deprived and unsafe, to one which became a place where residents felt safe and reported greater confidence in the justice system. The centre had become a 'prominent fixture in the Red Hook neighborhood' and 'arguably the best-known community court in the world' (Lee et al. 2013, p. 3). The core functions and broader aims of the court were to offer a 'dual commitment to changing the lives of individual offenders and the quality of life in communities' (Lee et al. 2013, p. 3). The court would deal with minor misdemeanours and offer alternatives to custody and fines, where defendants were strictly monitored to ensure compliance with their sentence. This sentence could include treatment for health issues such as addiction to drugs or alcohol, mental illness and could also include social services for other forms of support required. At the Red Hook Community Justice Centre, the judge would ensure that even the most minor offences faced a 'meaningful sanction' as soon as possible after leaving court, which would also offer a form of reparation, for the harms caused to the community, as part of the process of justice (Lee et al. 2013). This would seem to demonstrate effective outcomes in relation to reducing re-offending and impacting the local community and its residents in a positive way.

In 2005, the Red Hook model was piloted in north Liverpool, with a purpose-built centre to house the court and other services. The aims of the North Liverpool Community Justice Centre were to reduce low-level offending, anti-social behaviour and fear of crime, alongside increasing public confidence and victim satisfaction in the CJS (Llewellyn-Thomas and Prior 2007). In November 2006, the Government announced plans to launch 10 new Community Courts, across England and Wales, based on the model adopted in Salford, where the community court would make use of existing magistrates or combined court buildings. A timely and focused evaluation of the NLCJC (Mair and Millings 2011) paid attention to the ability for such initiatives to engage citizens and offer 'a unique court process with wider community resource provision' (p. 3).

In a community court, where problem solving approaches are adopted, the role of restorative practice is represented in a number of ways. Firstly, defendants are given an opportunity to address the problems leading to their offending and, by pleading guilty, taking responsibility for their behaviour. In addition, given that restorative justice promises to deliver 'reassurance that what happened was wrong, that something is being done about it, and that steps are being taken to discourage its recurrence' (Zehr 2002, p. 195), the problem-solving approach presents transparency in showing how the court is implementing measures to prevent re-offending and support victims (Mair and Millings 2011; Llewellyn-Thomas and Prior 2007). Community courts also aim to have a positive impact on the quality of life of the local community, and specifically to offer in house support services to deal with debt, housing issues and legal advice (Mair and Millings 2011). Again, in an alignment with restorative practice, this represents a more inclusive approach to the administration of justice, as it engages offenders, victims and the local community in the processes of justice and reparation. This shift requires the engagement of citizens to accept this as a form of justice in their community, to enhance feelings of 'connectedness' (McCold and Wachtel 1997, 2002).

The key features of problem-solving courts follow Berman and Feinblatt's (2015) model which requires monitoring of progress by the court, addressing the causes of crime as they relate to defendants' circumstances, working with others in partnership to address these problems and empowering offenders to engage in their rehabilitation. There is alignment here with the principles of restorative practice—accessibility and respect through the inclusion of the offender in determining their sentence, and voluntarism through this engagement in order to increase compliance and accountability (RJC 2016). Another

important component in restorative practice is to focus on offender rehabilitation and victims' rights in response to crime, as opposed to more punitive and retributive forms of justice. If done correctly, restorative justice can empower the victim and offender, giving them control over the nature of the reparation (Wright 2008; Johnstone 2013). Organisations such as UNITE offer expertise and forums for others to come together and deal with crime using restorative approaches. Studies have demonstrated a range of positive outcomes such as victim satisfaction (Shapland and Hall 2007), reduction in re-offending and a positive impact on community cohesion (Sherman and Strang 2007). King (2008) notes the contribution of restorative practice as representing a 'significant influence' (p. 19) on community courts. Furthermore, Walgrave (2011) suggests that in an adversarial system, the courtroom becomes a place in which clear communication suffers due to the threat of punishment, acting as a barrier to positive change. Restorative practice in courts offers an opportunity for 'authentic communication' (p. 117), aligning with problem solving approaches in court which advocate for dialogue between the defendant and judge to establish what has led to their offending.

However, there are legitimate concerns raised about restorative practice in the justice system. Restorative justice has been defined as a process whereby mediation occurs between victim and offender to repair harms caused by offending (Zehr 2002; Marshall 1999; Johnstone 2013). This somewhat simplistic definition masks the challenges apparent in truly understanding how restorative practice works in dealing with criminal cases, due to variations in practice, where and when it is applied and on the contradictory evidence on the outcomes of this approach (Cunneen and Goldson 2015). In addition, Lanterman (2021) suggests that while restorative justice requires voluntary participation, there are circumstances in which coercion into the process occurs. This is manifest as obligations of inclusion, raising the question about the efficacy of such encounters under these conditions to genuinely repair harms (Ward and Langlands 2008). Geeraets (2016) refers to the 'fiction' of voluntary participation, in that offenders are effectively shamed into taking part, which can impact the authenticity of their involvement and motivation to change.

It is also possible that defendants may accept an alternative approach to their case out of self-interest rather than a genuine acceptance of responsibility and accountability (Lanterman 2019). These concerns highlight the question of the authenticity and therefore, the potential effectiveness on such outcomes, to both change offender behaviour and support victims. While restoration, inclusion, accountability and problem solving present positive concepts in dealing with the harms caused by crime, it is clear that acceptance of their effectiveness is problematic and assumes their implementation is predominantly successful.

## 3. The Role of Therapeutic Jurisprudence in Community Courts

The contribution of therapeutic jurisprudence in the court setting reflects features which consolidate legal process and concerns about the welfare of defendants, victims and witnesses (King 2008; Ward 2014). In addition, it is suggested that this therapeutic effect extends to the community, where citizens work with the state to solve problems, and come together to repair harms caused by crime and anti-social behaviour (Lacey and Zedner 1995; Shapland 2008; Donoghue 2014). Ward (2014) has identified the growth of community courts, in the UK and other countries, as an important development in criminal justice policy, even though the courts evolved to be a different model from community justice centres in the USA. She emphasises that a key component of community courts is 'therapeutic jurisprudence' (p. 2). This is represented by court processes which enable offenders and others to develop different self-identities where they engage in lawful and purposeful activity, or 'a criminal justice model that has well-being at its core and puts a human face to the delivery of justice' (Ward 2014, p. 2). In the USA, Rottman and Casey (1999) chart the evolution of therapeutic jurisprudence and the problem-solving court model as stemming from courts being placed in the 'frontline of responses to substance abuse, family breakdown, and mental illness' (p. 13). The limitations of the court in addressing these problems became clear in the in the USA as manifest in views of the judiciary as

unresponsive, ineffective, and out of touch. In this context, therapeutic jurisprudence offered a framework which encompassed the principles of justice, law and concerns about the mental health of those going through the court system (Rottman and Casey 1999). Its primary aim is to enable legal processes to occur without causing detrimental outcomes to defendants' health, in direct contrast to adversarial systems of justice—crime becomes a 'series of problems to be solved' rather than a 'contest to be won' (Karp and Clear 2000, p. 328). This requires judges and magistrates to direct their attention to the needs and circumstances of defendants, beyond their duties as sentencer, where they will also be required to interact with defendants and their representatives in a different way. This individualised approach (Wolf 2007; Donoghue 2014) also requires signposting to services and therefore knowledge about local provision. A further important element is the use of the processes of the court as an opportunity for defendants to take responsibility for their behaviour and acknowledge the need for change.

King (2008) has referred to courtrooms as becoming 'dumping grounds' to address substance misuse, mental health problems, homelessness and other issues. In addition, the problems highlighted by Bowen and Whitehead (2013) concerning delays and the revolving door of the courts has led to widespread dissatisfaction with the court system and an increasing rationale for a different approach. In an analysis of specialist courts, Donoghue (2014) emphasized the need for training to enable magistrates to grasp the different processes of justice which aim to punish offenders and more effectively prevent re-offending, given the scepticism identified by Bowen and Whitehead (2013). However, even with the 'therapeutic' ethos, the focus on the causes of crime and evidence of the effectiveness of this approach (Ministry of Justice 2014), problem-solving approaches have remained on the periphery of the justice system. Research has examined the effectiveness of community courts in relation to outcomes such as a reduction in re-offending, improving the quality of life for the local community and increased compliance with court orders (Llewellyn-Thomas and Prior 2007; Mair and Millings 2011; Annison et al. 2013). This evidence has not persuaded central governments across the political spectrum or the senior judiciary to advocate for reform and change. This resistance to change remains, alongside a lack of investment as a legacy of austerity measures since 2010 and the impact of the COVID-19 pandemic. The impact of this is delays in criminal cases being heard in court, higher rates of discontinuation and lower rates of conviction, especially for sexual assault cases. The broader effect of this is erosion of trust and confidence in the criminal justice system (Victims' Commissioner 2021; House of Commons Committee of Public Accounts 2022). These problems are not easily solved without meaningful investment and political will to drive change and reform. The current state of the court system presents an opportunity to reflect on how justice can be delivered in more innovative and less harmful ways, which maintain the authority of the state.

## 4. Methodology

The data for this article draws from a wider study which explored the prominence of 'community' in criminal justice and social policy, as a concept to promise a more locally focused response to crime and aims to improve the overall quality of life for local residents. Using a case study of Middlesbrough, it focused on community justice initiatives, led by the police and courts, working with the third sector and other agencies to manage offenders and support victims. This included restorative justice practitioners and agencies working in partnership with them and those working in the community court, situated at the Teesside Combined Court building. The principal research tools adopted for this study were semi-structured interviews, which formed the focus for the data analysed for this article, to examine the contribution of restorative practice and therapeutic jurisprudence in the problem-solving approaches adopted by Teesside Community Court.

### 4.1. Sampling and Data Collection

Semi-structured interviews offered an opportunity to gain insight into the experiences of working in a community court, from those directly involved and others working in partnership with them. It was important to have a semi structured interview schedule to ensure a focus on common themes whilst allowing participants the freedom to express their experiences and views (King and Wincup 2008). This was important in the initial interviews which were exploratory in examining experiences of working in the community court, identifying participants through snowball sampling and establishing key themes for the focus of the research.

The sample for the wider study included 23 participants from the police service, courts, probation, restorative justice providers, Victim Support and local residents. For the purpose of this study, the data comes from magistrates working in the community court, policy staff working with the courts and police service, restorative justice practitioners and senior police officers who commented on the use of the problem-solving approach (*n* = 12). The data from these participants was re-visited to allow for a more focused examination of views on the features of the community court, which reflect the key principles of restorative practice and therapeutic jurisprudence. While in the original study, the participants were not directly asked about these concepts, it was clear in the process of re-examining this data and the conclusions drawn that participants had discussed and reflected on this, in the context of the practices adopted in the community court model.

### 4.2. Coding and Analysis

In the original study (Atherton 2020), thematic coding of the data was completed using qualitative analysis software (NVIVO). For this article, the process was repeated manually using data from the smaller data set with a specific focus on key concepts of restorative practice and therapeutic jurisprudence. Miles and Huberman (1994) describe codes as 'labels', 'tags' or 'categories' where varying amounts of text can be placed. These codes can then be organized to provide examples of themes being discussed in a study, much like a book index. Table 1 reflects this process, adapted from the original case study to include codes generated from interviews which focused on the role of restorative practice in the community, to deal with crime and disorder. This first category refers to the area of focus in terms of the groups interviewed (CJS staff and volunteers) and the key themes emerging when discussing experiences of working in the community court and views on this approach. For example, it was clear that 'diversion', 'problem-solving' and 'partnership working' were common issues discussed by participants across CJS staff and volunteers. Restorative practice was often discussed in the context of dealing with anti-social behaviour, young people and as providing support for victims.

**Table 1.** Sample of codes generated from interviews relating to restorative approaches.

| | Court Staff | Restorative Justice Workers | Police | Victim Support Staff |
|---|---|---|---|---|
| **Restorative practice** | Diversion<br>Sense of justice<br>Anti-social behaviour<br>Young people<br>Problem solving<br>Partnerships | Diversion<br>Anti-social behaviour<br>Logistical issues<br>Management support<br>Partnerships | Victim support<br>Sense of justice<br>Victim impact | Victim support<br>Offender's needs<br>Confronting behaviour<br>Repair harm<br>Sense of justice<br>Anti-social behaviour<br>Young people<br>Problem solving<br>Partnerships |

Table 2 shows how this process of coding identified features relating to therapeutic jurisprudence within the interview data, when participants discussed their experiences and views of the community court. Common themes included the importance of dialogue with defendants, shifting away from legal traditions and the need for a different approach.

**Table 2.** Coding frame reflecting the re-examination of data from participants working in and with the community court.

| | Court Staff | Restorative Justice Workers | Police | Victim Support Staff |
|---|---|---|---|---|
| **Therapeutic jurisprudence** | Dialogue with defendant<br>Legal traditions<br>Engagement in process<br>Intervention<br>Revolving door<br>Public perception<br>Mental health impact<br>Defendant behaviour<br>Formal setting | Inclusion of offenders<br>Empowering defendants<br>Engaging citizens | Different approach<br>Young people at risk<br>Public perception<br>Logistical challenges | Better outcomes<br>Engagement in process<br>Motivation for change |

## 5. Findings from Qualitative Data

The findings presented here represent the views and experiences from practitioners working within the courts, police service and those working in restorative practice and the third sector. As stated above, the data from the original study was re-examined and analysed to identify themes relating specifically to restorative practice and therapeutic jurisprudence emerging from participants' accounts. The findings are presented to align with the codes identified above, drawing directly from interview data in the form of common themes discussed and direct quotes as examples of this.

### 5.1. Therapeutic Jurisprudence and Restorative Practice in Court

The Teesside Community Court followed the problem-solving model which allowed magistrates to seek information about individual defendants, to tailor their sentence to address the problems leading to their offending. Services would be signposted by court support workers, to engage defendants with help and also as part of ensuring they took responsibility for their offending and for their rehabilitation. A key part of this process for the defendant was that they needed to enter a guilty plea, and would be offered a different approach (Wolf 2007; Donoghue 2014). For the judiciary, this required abiding by a new set of principles so that when cases came court, there was a genuine attempt to prevent re-offending:

> *So, yes, the problem-solving, that came about when I first took over, we had a health check, it was a set of principles . . . .we looked at people who were coming into court, either for the first time or coming back for low-level offences and we decided it might be a good idea to provide some intervention at that point, to stop that revolving door or to stop people progressing to more serious crimes. (Police/Community Justice Liaison)*

This reference to a 'health check' creates a different form of interaction in court, presenting a key feature of therapeutic jurisprudence to provide an opportunity for defendants to express their needs and for the court staff to respond to this (King 2008; Ward 2014). Many participants working with the community court referred to its value in addressing low-level offending and anti-social behaviour, where there could be some meaningful intervention to prevent escalation of offending and improve the quality of life in the community (Mair and Millings 2011; Donoghue 2014; Ward 2014). In these accounts is a clear reflection of what many felt was an important change in the delivery of justice in the community, in addressing the problems leading to crime and removing this from the sole provision of the state (Christie 1977). The community court was seen as a place where staff could find the triggers for offending, with the understanding that this behaviour was the culmination of a series of problems, and required insight into the circumstances and experiences of defendants:

> *We would ask [defendants] to have a word with the problem solver in court, who would take them through a list of questions because you realise that everybody who offends, there*

*is some kind of trigger for it even if it's you know, absentmindness through stress at work or stress at home and they have shoplifted and they did not mean to . . . so we would take them through this list of criteria to find out what the triggers are. (Magistrate in the Community Court 1)*

Along with revealing individual 'triggers' for offending, it was clear this approach would attempt to engage with other agencies to deliver solutions to problems, and to also understand more about the community in which defendants lived. This further aligns with the ethos of focusing on the well-being of defendants when determining the response to their offending (Rottman and Casey 1999; King 2008). Allowing them to explain their behaviour and what led them to offend means court staff would be able to respond with a sentence and support plan to achieve a more effective outcome. This would necessitate working in partnership with others, for example, advice surgeries based an area of Middlesbrough, North Ormesby. This formed an important aspect of the work of the court, helping defendants address issues such as debt, housing problems and health concerns. It was therefore vital that both magistrates and defendants were prepared to engage in a dialogue in the courtroom, in order to facilitate access to support:

*It was not always about sentencing and punishing, it was about getting them help and back into the community, which I think is important. I do have my community justice head on when I am in court, I think it really does work. (Magistrate in the Community Court 1)*

Advocates of therapeutic jurisprudence emphasise the need for legal processes to be conducted with defendants' well-being in mind, in order to engage them in the process and encourage them to comply with court orders (Rottman and Casey 1999; King 2008; Ward 2014). It was reported that when offenders were offered support and welfare, many took the opportunity, but they also valued just being listened to:

*I think just having somebody who would listen to them to be honest, you know they said there is nobody in their life has listened to them, this might be the first time in court and they have issues, money, drugs, alcohol; there is a whole range of things. (Magistrate in the Community Court 2)*

This process of dialogue and listening to the offender in the formal setting of the court clearly reflects the ethos of therapeutic jurisprudence, and emphasises how the courtroom can be used as a setting in which to begin the process of reparation. Restorative practice is also evident here, in the form of creating a situation which moves the defendant toward taking responsibility for their actions (Johnstone 2013). The Red Hook model demonstrated the importance of monitoring and ensuring compliance to ensure confidence in this approach was maintained (Donoghue 2014), while giving defendants the opportunity to demonstrate change. This need for allowing time for compliance to be demonstrated was clear in one case:

*I had one lad, the first time I've seen him, he could hardly stand up he was that drunk, I followed him through and could gradually see an improvement . . . he would not talk to you at first but he then he would, about what he had been doing. (Police/Community Justice Liaison)*

This change in the defendant's demeanour indicates the impact of a different approach, eventually leading to him engaging with the court staff and taking responsibility for his behaviour, a key component of restorative practice (Marshall 1999; Sullivan and Tift 2001; Johnstone 2013). In a traditional court, such a case would ring alarm bells about the issue of this defendants' behaviour in court, but the difference in approach acknowledged his problems stemmed from alcohol misuse. This was possible as the judge engaged in a dialogue with the defendant, persisting with this while monitoring progress and making allowances for their conduct in the initial hearings (Ward 2014).

The goals of restorative practice are to preventing re-offending and enable re-integration for offenders, alongside supporting victims (Zehr 2002; Marshall 1999). Practitioners work-

ing with UNITE reported on the importance of a problem solving and restorative approach, specifically mediation within a young offender's family:

> *We will also work with the youth offending team, the young offender and their parents and to give them support to stop them re-offending. The focus is on problem-solving, conflict management skills, try and help them to help themselves. (Restorative Justice Mediator)*

Working in partnership with others is not distinct to the community court, as it has been enshrined in policy and legislation since the introduction of the Crime and Disorder Act (1998). In this instance, there was a clear goal of resolving conflict which had arisen within a family, which was identified in the courts as a problem which could be addressed through mediation. From those working in restorative justice, there was also an aim to empower families to resolve problems themselves and prevent conflict.

Those working in the community court saw an opportunity to adopt a restorative justice approach more widely:

> *More use of restorative justice is helpful the courts can make this part of an order, it is not something always before court. We are about to start neighbourhood justice panels, there are pilots out there. There are concerns if it takes work out of the courts, but again it gives the community the chance to take a bit of control it could work out well. (Magistrate in the Community Court 2)*

It is interesting to see here reference to taking cases out of the work of the courts, and back to the community, where neighbourhood justice panels represent a process, which is placed in the hands of local residents and local state agencies. The placing of control back to the community is clearly important for victims of crime (Christie 1977), but also for other residents to see justice as a force for unity and solidarity (Rawls 1971), rather than the sole responsibility of the state. This does firmly place restorative justice as part of the court response to offending, although setting the parameters for this in a place which traditionally metes out punishment could be problematic (Cunneen and Goldson 2015; Lanterman 2021).

In addition, the organisational culture and practices of the judiciary were also cited by one of the community court magistrates as presenting potential barriers to effective outcomes, reflecting the language and demeanour of judges:

> *Yes it can have so many different ways of presenting itself and how . . . it can be difficult because it can be sterile and so fixed in our ways, the way we have done it for 100s of years. Even the terminology, things we do not think are terminology such as 'standing a case down' which for us is easy, we are just going to put it off for half an hour, or we are going to 'adjourn' and terms like 'bail'. So we really need to step back and think about these things and how people . . . even nerves can cause difficult behaviour. (Magistrate in the Community Court 1)*

This consideration of the formal setting of the court has shown how adopting problem-solving approaches that put a 'human' face to justice align to the principles of therapeutic jurisprudence, shifting from adversarial models (Ward 2014). Research by Annison et al. (2013) also revealed the 'feel good factors' and 'pioneering spirit' of problem-solving courts, as achievable even in a formal setting with clear legal requirements. This perhaps does not shift completely away from legitimate concerns about coercion into different approaches to deal with criminal cases, but it does allow for less punitive and stigmatising processes to occur. The 'unequivocally positive idealisation' (Cunneen and Goldson 2015, p. 4) which has been presented by those advocating for restorative justice needs to be understood in the context of the legal requirements of courts and the system of laws which must be adhered to. The reference to problem solving approaches as 'alternative justice' needs to be understood on the context that this all occurs in a formal courtroom setting, which enables monitoring of compliance with sentence conditions. For some this may present as a form of coercion into a process of justice, but it is important to consider that from the victims' and

local community's perspective, justice still needs to be seen to be done (Mair and Millings 2011; Donoghue 2014).

The practices found in the community court in Teesside demonstrated both the need for and value of taking a different approach. By adopting elements of restorative practice and therapeutic jurisprudence, the court becomes a place in which meaningful change for defendants can occur, as opposed to remaining a 'dumping ground' for a range of social issues which lead to offending (King 2008). The uniqueness of this approach can have a wider impact on the local community (Mair and Millings 2011) and provide a means by which to reform and improve the effectiveness of courts (Bowen and Whitehead 2013).

*5.2. Challenges to Implementing Change*

The participants involved in this study expressed positive outcomes from working within the community court; however, many also referred to challenges when implementing change in the delivery of justice. For example, while some magistrates saw the use of a 'problem solving checklist' as helpful in building a rapport with defendants, others referred to resistance to such an approach:

> *Yes, we always promoted direct engagement, we did quite a lot of training with magistrates as well around direct engagement, when we rolled out further and some of them were horrified at the thought of speaking to an offender. The culture for them is that this is done by the advocates, they find it incredibly difficult. Others are absolutely brilliant, they can make a huge amount of difference to the way that things go, because they build a rapport and can challenge them about their behaviour. (Police/Community Justice Liaison)*

This account of resistance is a concern, given the importance of judges engaging with defendants in a dialogue both to tailor sentence to individual needs (Wolf 2007; Donoghue 2014) and to empower defendants to take responsibility for their behaviour. The practices of the judiciary were cited by one of the community court magistrates as creating a barrier with the use of legal terminology. but also, at times, the demeanour of judges towards defendants:

> *Yes it can have so many different ways of presenting itself and how . . . it can be difficult because it can be sterile and so fixed in our ways, the way we have done it for 100s of years . . . the terminology such as 'standing a case down' which for us is easy, we are just going to put it off for half an hour, or we are going to 'adjourn' and terms like 'bail'. So we really need to step back and think about these things and how people . . . even nerves can cause difficult behaviour. (Magistrate in the Community Court 1)*

This reflection of the impact of the formal setting and processes of the court has demonstrated the need to consider problem-solving approaches. It further emphasises that community courts can put a 'human' face to justice, enabling a shift from adversarial models which can create divisions and conflict (Ward 2014). It presents an opportunity for justice to become a means by which to reinforce the 'bonds of civic friendship' (Rawls 1971, p. 5), moving away from a focus on deterrence and vindicating the punitive functions of the CJS.

Across the sample, participants referred to the challenge of informing the public and having them accept a different way to deal with crime and anti-social behaviour. It was clear this was in part about explaining the work of the community court, or how restorative justice works and also counteracting misleading media representations. The message court staff felt was most important was that punishment of offending remained at the heart of their work, but there was a way to do this differently:

> *[Community courts] I think, are misunderstood and we need to tell the public, as when they hear about a case like an assault, this makes the headlines, so we need to explain our work . . . My role as a magistrate is not just to punish, but to punish in the right way so the community benefit from this, it's easier to send someone to prison, harder to address the problems. (Magistrate in the Community Court 1)*

The emphasis on punishment is important here, perhaps reflecting the view of the public as 'punitive', but certainly to appeal to governments who seek law and order as a mechanism to assert their authority and to be seen as effective in reducing crime. Without this, alternative approaches to crime are likely to remain as a peripheral response, even with the evidence of the need for reform (HoCJC 2010; Bowen and Whitehead 2013).

With reference to the role of the judiciary in this process, Mair et al. (2007) emphasise that if magistrates were more aware of the options and if community provisions to support offenders' desistance were properly resourced, this would aid in the effectiveness of community sanctions. However, among participants there was often reference to the lack of support from senior management:

> *I still think that at the more senior officer level, if there is not the commitment there it is never going to happen. Because they are the ones ultimately that can say yes we can do this, we cannot and I think there is a lack of understanding about some of the changes at the top. They do not really understand what community justice is about so therefore it's not a priority for them. (Police/Community Courts Liaison)*

There is a clear concern here about the view that community justice initiatives are not prioritised and are mis-represented, despite evidence from cross party inquiries into the need for reforming the CJS and getting the message across to citizens of the value of such changes (HoCJC 2010). Those working in the community court also cited the need for local media to support change:

> *I can see both sides, my background is journalism, I have done court reporting and I know you are looking for the hook, the punchy intro and something that draws the public in and reaches out to them, and I know the legal guidelines. But it is very difficult to explain to the public how and why we do what we do, you cannot put it over in an easy sentence so although we work well with the media, one damaging headline can undo years and years of work. (Magistrate Community Court 1)*

These damaging headlines contribute to residents' fear of crime and view of the justice system as a means to enact punishment of offenders (Garland 2001). It is important to acknowledge the role of the media, given the focus on this issue in the HoCJC (2010) report, with reference to the need to better inform the public about the CJS. The reporting of crime, along with the lack of senior management support, meaningful investment and the acknowledge of the role of wider inequalities all act to impede the implementation of 'alternative justice' (Cohen 1985; Garland 2001). This does not undermine the principles of restorative justice or the aims of problem-solving courts, but does emphasise the current limitations of their role in mainstream criminal justice practice. In addition, if justice has a part to play in stabilising communities (Rawls 1971) and improving the quality of life for local residents, this would seem to advocate for a shift from punitive responses to crime, towards the ethos of restorative practice and therapeutic jurisprudence.

Participants referred to a desire to adopt restorative justice, but also with some recognition of the challenges in implementation. If restorative justice is meant to repair harms and discourage reoccurrence of offending (Zehr 2002), it is a concern that there are logistical challenges and also a lack of motivation to implement it. This is further emphasised below with reference to the lack of will to go beyond talking about this new innovation:

> *I think that restorative justice is great, but I do think a lot of agencies use it as sort of a buzzword. You know they say yes we support it and believe it's the right way forward, however my experience of it is very different, because with all those people selected, by those agencies, the difficulty then was trying to get people to take time out from their workplace and deal with these cases. (Victim Support Volunteer 2)*

There is clearly also a need to consider how well the community is set up to facilitate restorative justice, so that it can provide practical support and 'peace-making criminology' to resolve conflicts (Quinney 2000; Sullivan and Tift 2001). As well as these logistical issues, those working with victims also emphasised a key challenge for the implementation of

restorative practice was the requirement that offenders themselves had to engage in the process:

> *It's still controlled . . . if the perpetrator does not want to take part in it, it cannot happen, if it was mandatory so they had to take part in it as part of their sentence, it would happen. But with restorative justice, the principle is, the perpetrator has to agree to it. If they do not, despite the fact the victim may feel it's a means to have their voice heard and they are quite excited as it's good for them, still it is controlled by the perpetrator, if they do not want to engage, it does not go ahead. (Victim Support Volunteer 1)*

This demonstrates the conflict presented by the legal context of using restorative practice in the CJS, where such processes are suggested to be part of a court order, which contradicts the principle of voluntarism (RJC 2016). Similar concerns were raised when referring to experiences with victims who reject the use of restorative justice. The capacity for such a response to be accepted when formal social controls and retributive processes are on offer could be undermined, depending on individuals' expectations of the courts and the justice system (Nellis 2000).

## 6. Discussion

In a system of justice criticised for its lack of focus on the reintegration of offenders (Nellis 2000) which also implements authoritarian measures which address citizens' fears (Garland 2001), the frustrations felt by practitioners who have witnessed the effect of new approaches such as community courts are clear. They offered an opportunity to provide a setting and legal framework by which to develop a more humane and effective response to crime. Instead, practitioners, researchers and others who have understood the potential of community courts have seen time and time again, how easily such innovations are side-lined and de-legitimised as both effective and as a form of justice. The support and rationale for adopting a problem-solving approach in the Teesside Combined Court was clear among participants from professional roles in the CJS. It was seen as a more effective way to address causes of crime, and this approach also highlighted the limitations of the CJS.

If such an approach is to be transformative and 'socially meaningful' (Donoghue 2014, p. 141), this will require effectively addressing low-level crimes and anti-social behaviour, through repairing harms, re-integrating offenders and supporting victims. The focus on problem solving approaches as used in the community court was important in order to understand the contribution of restorative practice and therapeutic jurisprudence. It was clear from the study that many practitioners understand that the CJS needs to be able to do more than respond to crime as a violation of law or to attempt to deter others (Nellis 2000).

Problem solving approaches used in courts offer a new way to do justice, in a setting which where a fairer and needs focused response to crime can occur. In addition, courts represent a place of authority, a setting where compliance and monitoring of offenders can take place, as an important part of the justice process (Bowen and Whitehead 2013). These benefits were embraced by participants working in and with the courts, where a shift in organisational culture was possible, albeit with some challenges. For example, therapeutic jurisprudence advocates for judges to engage in dialogue with defendants, which was seen as a very important and positive aspect among magistrates (Ward 2014). While there are clearly challenges, it was clear that both therapeutic jurisprudence and restorative practice contribute to the core principles of community courts, which were identified as important elements by participants. This is manifest in understanding how the circumstances of defendants should inform sentencers, who should then have an opportunity to monitor progress to increase accountability. Such an approach will also require collaboration with other agencies to address defendants' needs (Ward 2014; Donoghue 2014). These aims align well with the principles of restorative justice, which require the use of a setting characterised by respect, accessibility, neutrality and safety (RJC 2016).

This model also demands a shift from adversarial contests in court towards creating a place in which defendants are empowered to be part of their rehabilitation, working with

the court staff and others to achieve this. This is a key feature of therapeutic jurisprudence, where legal processes are implemented with a focus on the impact of these and the welfare of defendants (King 2008; Ward 2014). This can also create a situation in which defendants see themselves as part of a community rather than as a burden to it, leading to the process of justice becoming more 'socially meaningful' (Donoghue 2014, p. 141). Engaging with policies which were not retributive but reflected restorative approaches to dealing with offenders was embraced by those practitioners who had been involved in the process and seen positive results. These accounts of effectiveness were also presented by those speaking on behalf of victims of crime. However, the processes of reparation and restoration were subject to media representations which were distorted by a lack of context and continuing calls for retributive justice. Media reporting was viewed as an obstacle to increasing participation in crime and justice issues on a local level, through irresponsible reporting which contributes to existing suspicions and fears among citizens (Cohen 1985; Jewkes 2011). This would reinforce views that participation in addressing the needs of victims and offenders poses a risk to safety and should remain as the responsibility of the state (Nellis 2000; Hough and Roberts 2004).

The problem-solving approaches of the courts were valued as a way to reduce re-offending, and it was important that this included monitoring compliance with a court order. This presents an example of the reciprocal relationship between defendants and the courts, to understand their role and responsibilities in achieving and maintaining desistance. It also provided a way for magistrates to be clear to residents that this approach was implemented within the legal requirements of the justice system, and that there would be consequences for non-compliance. This could arguably be seen as coercion and a more authoritarian process distinct from restorative practice values, but it is important to reassure local residents and others that this approach has the potential to be both an effective way to deal with crime and one which holds offenders to account.

There was evidence of the transformative effect of community courts to deal with persistent problems and establish the local community as a safe place to live. The challenges highlighted must not deflect from both the need for change and also the potential of a therapeutic ethos and restorative practice in our legal system to offer a starting point for justice to become transformative. This study took place during the implementation of the community court model; however, as the research continued, it was clear this investment was not going to be sustained. Once again, justice being done in a community setting was to remain on the periphery of the CJS, given a political ideology which reinforced justice as punishment, deterrence and a system which views crime as a violation requiring a punitive and repressive response.

Further research needs to continue to focus on the impact of broader socio-economic conditions and political ideologies which inform policy and practice to deal with crime and anti-social behaviour. There also needs to be continued focus on the public understanding and experiences of justice, and how this affects views about the responsibilities and expectations of both citizens and the state. This research has examined and highlighted some of the issues which face policy makers and practitioners attempting to find new and more effective ways to do justice and improve the quality of life and safety in communities. These challenges demonstrate a need for a shift from a justice system which divides communities and which is limited to crime control measures. The therapeutic and restorative qualities of community courts and other forms of alternative justice demonstrate that crime and anti-social behaviour are indeed problems which can be solved, thereby having a transformative effect on victims, offenders and communities.

**Funding:** This research received no external funding.

**Institutional Review Board Statement:** The study was conducted in accordance with the Declaration of Helsinki, and approved by the Research Ethics Committee of De Montfort University (protocol code 886 and date of approval: 19 April 2012).

**Informed Consent Statement:** Informed consent was obtained from all subjects involved in the study.

**Data Availability Statement:** This article used data gathered from a PhD study, some of which is available in the final thesis online (Susie Atherton PhD Thesis Sept 2018.pdf (dmu.ac.uk)).

**Conflicts of Interest:** The author declares no conflict of interest.

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
