# Peer review of "Restorative Practice and Therapeutic Jurisprudence in Court: A Case Study of Teesside Community Court"

_laws, 2022_

Round 1

Reviewer 1 Report

This is an interesting paper which could contribute to thee extant literature provided that a number of revisions are made.

The paper is based on a small qualitative study that was carried out in England looking at the implementation of restorative justice and what can be learned from it for future practice and policy.

The main issue with the paper is the lack of clarity around 3 concepts: Community justice, restorative justice and therapeutic jurisprudence. These are three different disciplines that need to be treated in their own right. This confusion impacts on the validity and clarity of the paper's main contribution i.e. the learnings from the interviews. For example, it is not clear whether the findings relate to restorative justice or community justice. Restorative justice is not community justice and it appears from the statements and the paper's main line of argument that the focus here is community justice.

I recommend that the authors review the notion of restorative justice in a more in-depth manner so that they can differentiate its substance, underlying values and approach from community justice. Philosophical and theoretical papers on restorative justice should help in helping them identify these principles and ethos.

The authors might also want to review their title and abstract and simply put emphasis on the role of community justice vs restorative justice. This should give a better sense of what the  main contribution of the research is.  

The case study is good but again it is not well connected to the main argument. The methodology here is also dubious. Other methodological questions relate to the sampling strategy for the interviews, original sample, response rate and main demographics of the participants. This is useful information to have given how small the scale of the fieldwork was. There are also similar studies focusing on evaluations of restorative justice in other areas in the UK such as London (e.g. see recent paper in Victims and Offenders on a London case study). These should help to connect past literature with the study's findings.

Some of the results are repetitive of findings from previous studies, which is not an issue per se. However, they do need to be linked with the main points of the paper so that they can add value. The argument on top down structures and support is an important one especially if put in the context of restorative justice which aims to bring these structures down. I recommend that the authors read papers on top down structures, power abuse, power structures and controversies surroundings the restorative justice movements and ethos.

Finally, the paper needs some editing as some of the points it makes are repetitive. It reads as if this is a shorter version of a longer report that was not very well edited. 

Author Response

Thank you for your comments, they helped me to re-focus the themes of the article to examine the contribution of restorative practice and therapeutic jurisprudence to the community court model. I have also addressed the methodology, again to make it more focused on how data was used for this article, from the original study and I hope this is now clearer. The review process allowed me to tighten up the work, making it more focused and concise and so I appreciate the time you took to provide this feedback - it really helped me consolidate what I was trying to get across in the original submission.

Reviewer 2 Report

General Comments:

Given the worldwide expansion of problem-solving courts and restorative practices, this manuscript deals with a timely topic, that is, the relationship between problem-solving courts and restorative practices. Therefore, this manuscript has the potential to contribute to the literature.

However, this manuscript also has four major challenges. First, the conceptual and theoretical relationship between problem-solving courts and restorative practices needs to be probed more deeply. This is partly because, in my view, problem-solving courts align more with therapeutic jurisprudence than with restorative practices. Indeed, while there are noticeable similarities, Daly and Marchetti (2012) described therapeutic jurisprudence and restorative justice as separate innovative justice responses with the former being more court-oriented and the latter being lay-person oriented. Given this distinction, the author(s) should explain how restorative justice as a concept (and/or practice) is compatible with problem-solving courts.

Relatedly, the author(s) should clearly conceptualise restorative justice. The definition, values, and process of restorative justice have long been contested. Some limited restorative justice to a face-to-face dialogue process, while others contended that restorative justice is the outcome we aspire to achieve. In places, the author(s) ostensibly described restorative justice as one agreed-upon concept (e.g., ‘It again also demonstrates the ethos of restorative practice’ on Page 13).

The relevance of section 2 is not clear. Specifically, this literature review section should be written in a way that there is a need for this current research. As it stands, only seemingly relevant literature is reviewed. Accordingly, neither the research gap in the literature (including research questions) nor the significance of this current research is clear.

Methodology and Results sections need more work. For example, it is not clear why Table 1 is presented and how it is used in the analytical process. In addition, sub-headings are confusing when common issues discussed by participants across CJS staff and volunteers (Page 11) were ‘problem-solving’,’ partnership working’, and ‘communicating change’. The author(s) should explain how the themes were identified in the analysis.

Specific Comments:

Page 7, ‘G’ in ‘in November 2006, the Government’ is bold, so it needs to be fixed.

Author Response

Thank you very much for your comments and feedback, I agree that I had not adequately linked literature to the data analysis and discussion, and this was a consequence of trying to include too many concepts - your suggestion to focus on therapeutic jurisprudence was very helpful and it helped me to also support the discussion on the inclusion of restorative practice. I amended the methodology to reflect the analysis for this work, and so I hope this is clearer. Obviously this had an impact on the discussion, which has focused on the key concepts and also reflected some of the challenges of implementing innovations in justice. 

Round 2

Reviewer 1 Report

This is an improved version of the paper but some key issues remain. These were highlighted in the previous round of review. My concerns regarding the lack of critical reflection of the extant literature remains. It is clear that the literature review that was carried out omitted recent work that has been published in the area. For example, the top down delivery of restorative justice and its associated concerns has been the debate of 2020 and 2021 books and articles on power. These are not included and neither the more recent debate on how to reconciled the community led vs the top down delivery of restorative justice. Various scholars have presented consensual models of delivering restorative justice which again are not critically evaluated by the authors. 

Author Response

Thank you once again for your comments and insight; I have as advised engaged with some critique of restorative justice around the issue of coercion/consent into the process and how this impacts the authenticity of the encounter and practice that aligns with community courts. I have also address spelling errors and grammatical issues and added some further clarification to the methodology.

Reviewer 2 Report

Thank you for your revision and reply. I can see the significant improvement in the quality of the manuscript.

Before publication, the author needs to address one challenge in the manuscript. That is, it is not clear why the sub-section '2.1 The role of therapeutic jurisprudence' comes under the section '2. The problem-solving court model and restorative practice'. How are they related to each other?

Author Response

Thank you once again for your comments, I am pleased you have seen the improvment; I have added some critical analysis on RJ and how this impacts the community court and some clarification in the methodology. I have also addressed some issues in the findings section to avoid repitition and present a clearer analysis.